# Molecular Mechanisms Underlying Pluripotency and Self-Renewal of Embryonic Stem Cells

**DOI:** 10.3390/ijms24098386

**Published:** 2023-05-07

**Authors:** Fahimeh Varzideh, Jessica Gambardella, Urna Kansakar, Stanislovas S. Jankauskas, Gaetano Santulli

**Affiliations:** 1Department of Medicine (Division of Cardiology), Wilf Family Cardiovascular Research Institute, Einstein Institute for Aging Research, Institute for Neuroimmunology and Inflammation (*INI*), Albert Einstein College of Medicine, New York, NY 10461, USA; 2Department of Molecular Pharmacology, Einstein-Mount Sinai Diabetes Research Center (*ES-DRC*), Fleischer Institute for Diabetes and Metabolism (*FIDAM*), Albert Einstein College of Medicine, New York, NY 10461, USA

**Keywords:** CRISPR/Cas9, embryonic stem cell (ESC), epigenetics, histone modifications, human ESC (hESC) vs. mouse ESC (mESC), Klf4, naïve vs. primed pluripotency, Oct4, pluripotent stem cells (PSC), self-renewal

## Abstract

Embryonic stem cells (ESCs) are derived from the inner cell mass (ICM) of the blastocyst. ESCs have two distinctive properties: ability to proliferate indefinitely, a feature referred as “self-renewal”, and to differentiate into different cell types, a peculiar characteristic known as “pluripotency”. Self-renewal and pluripotency of ESCs are finely orchestrated by precise external and internal networks including epigenetic modifications, transcription factors, signaling pathways, and histone modifications. In this systematic review, we examine the main molecular mechanisms that sustain self-renewal and pluripotency in both murine and human ESCs. Moreover, we discuss the latest literature on human naïve pluripotency.

## 1. Introduction

The term pluripotency indicates the ability to generate cells from all three embryonic germ layers. Pluripotent stem cells (PSCs) are cells with an unlimited self-renewal potential that are able to differentiate into virtually every type of cell: PSCs can maintain their pluripotent state indefinitely, thereby providing a suitable model to study the mechanisms of self-renewal [1,2,3].

PSCs are derived from different stages of early embryonic development and maintain self-renewal potential in vitro using exogenous cues. There are distinctive types of PSCs, including embryonic stem cells (ESCs) and induced PSC (iPSCs): ESCs are derived from the inner cell mass (ICM) of human blastocysts (known as human ESCs, hESCs) [4,5,6] or from the ICM of the mouse pre-implantation blastocyst (known as murine ESCs, mESCs) [7,8,9,10], whereas hiPSCs are obtained from somatic cells such as dermal fibroblasts and blood cells following the activation of certain reprograming factors, including Octamer-binding protein 4 (Oct4, also known as POU5F1: POU Class 5 Homeobox 1), Nanog, Krüppel-like factor 4 (Klf4), and c-Myc [11,12,13,14,15,16,17,18].

PSCs can produce three germ layers in vitro, including endoderm, mesoderm, and ectoderm. Self-renewal is associated with cell cycle control, and there are unique mechanisms controlling the cell cycle of PSCs; these mechanisms support unlimited proliferation and capacity to differentiate (Figure 1); discriminating between self-renewal and differentiation primarily depends on the expression of specific pluripotency markers [19,20,21,22,23,24,25,26,27,28,29].

PSCs represent a tremendous tool for cell therapy, regenerative medicine, disease modeling, and drug discovery [30,31,32,33]. Several studies have shown that the pluripotent state and differentiation suppression of ESCs are regulated by core stemness transcription factors (Oct4, Nanog, and Sox2), extracellular signaling pathways, nucleus transcriptional programs, chromatin remodeling, and epigenetic modulators [21,34]. In ESCs, gene regulator networks driven by master transcription factors play a key role in pluripotency maintenance. Stemness transcription factors promote the expression of genes related to pluripotency and self-renewal, thus supporting ESC pluripotency. Understanding the molecular mechanisms underlying self-renewal and pluripotency of ESCs is vital for clinical applications.

Pluripotency is regulated by specific factors in mESCs and hESCs. Indeed, mESCs show a “naïve” state of pluripotency that is mainly dependent on Leukemia inhibitory factor (LIF) and bone morphogenetic protein-4 (BMP4) signaling, while hESCs represent a “primed” state of pluripotency which is relatively similar to the murine epiblast stem cells derived from the post-implantation blastocyst [35,36,37,38,39,40,41,42,43]. In this review, we summarize different signaling pathways and epigenetic modifiers involved in the control of pluripotency state and self-renewal of PSCs in both mESCs and hESCs.

## 2. Murine ESCs: Signaling Pathways Supporting Self-Renewal and Pluripotency

The members of the TGF-β superfamily, including TGF-β, BMP, Nodal, and Activin, are fundamental for the maintenance of pluripotency and self-renewal of stem cells during development [44,45,46]. LIF and BMP4mare two important molecular factors promoting the self-renewal capacities of mESCs. Specifically, LIF sustains the self-renewal of mESCs by activating transcription factor STAT3 (signal transducers and activators of transcription 3); however, LIF alone is insufficient to prevent neural differentiation and support self-renewal of mESCs in culture, and combination with BMP is necessary to properly induce inhibitory differentiation (Id) genes through the “Mothers Against Decapentaplegic Homolog” (Smad) pathway [47]. 

BMP sustains pluripotency in mESCs by maintaining an undifferentiated state through the suppression of the extracellular signal-regulated kinase ERK/MAPK pathway [48] and the activation of MEK5/ERK5 [49]. Notably, BMP signaling can induce mesodermal fate even in absence of LIF [50] (Figure 2).

The transcription factor Nanog has been shown to protect self-renewal potential of ESCs even in absence of LIF, and is generally considered a master regulator of pluripotency [51,52]. Nanog induces an undifferentiated state by repressing Gata4 and upregulating Rex1 and LIF-responsive genes, in particular Esrrb [52,53,54,55]. Oct4 is another central factor that can sustain intrinsic signaling in a LIF-independent manner to promote ES cell pluripotency and self-renewal [56,57]. Natriuretic peptide receptor A (NPR-A) upregulates gene expression of pluripotency markers (Nanog, Oct4, and Sox2) and increases levels of phosphorylated Akt, hence promoting self-renewal and pluripotency of ESCs [58]. Additionally, augmented intracellular AMP or AMP/ATP levels activate adenosine monophosphate-activated protein kinase (AMPK) signaling, which plays pivotal roles in differentiation, cell cycle, metabolism, and self-renewal of stem cells [59,60]. Indeed, Alba and co-workers demonstrated that 5-aminoimidazole-4-carboxamide ribonucleoside (AICAR), an AMPK activator, promotes stemness and self-renewal properties in mESCs via the PI3K/GSK3β regulation of Nanog expression [61].

Smad7 is a well-established negative regulator of the TGF-β/Smad signaling cascade [62]. Nevertheless, Yu and collaborators indicated that Smad7 is able to promote ESCs self-renewal in a TGF-β independent manner, by activating STAT3 signaling [63]. Epidermal growth factor receptor (EGFR) is an upstream regulator of STAT3 phosphorylation in cancer stem cells [64,65], and therefore has critical roles in proliferation, differentiation, and apoptosis [66]; specifically, EGFR inhibition in mESCs significantly reduces cell proliferation and gene expression of pluripotency markers, disrupts embryonic body (EB) formation, and upregulates differentiation genes (Gata4). These findings demonstrate the relevance of EGFR in self-renewal and pluripotency of mESCs [67]. A recent study has shown that silencing the histidine phosphatase properties of the phospholysine phosphohistidine inorganic pyrophosphate phosphatase (LHPP) promotes self-renewal of mESCs via positively regulating the Wnt/β-Catenin pathway [68]. In addition, inhibition of PKC by CID755673 maintained pluripotency via increasing the level of AKT phosphorylation and activation of PI3K/AKT signaling [69].

Equally important, Park and colleagues reported that the knockdown of melanoma-associated antigen D1 (Maged1) causes an increase in ectodermal differentiation and reduction in ERK1/2 phosphorylation, cell cycle progression, and teratoma formation in nude mice [70]. Maged1 has been shown to be involved in many biological processes, including proliferation, differentiation, and maintenance of mESCs self-renewal [71,72].

The tumor repressor p53 gene induces cell apoptosis, senescence, and cell cycle arrest in stressed somatic cells to prevent the transmission of genetic abnormalities to their offspring [73,74]. Moreover, p53 is critical for the genomic stability of hESCs and iPSCs [75]. Chromatin immunoprecipitation (ChIP) analyses have revealed that p53 induces expression of differentiation genes and suppresses pluripotency genes [76,77]. Therefore, it has been speculated that p53 could exclude ESCs with DNA damage from a pluripotency state by repressing Nanog expression [75]. Various studies have shown that nucleolar proteins are also effective in proliferation and survival of stem cells. For instance, Nucleolin (NCL) maintains mESCs self-renewal via repression of the p53 Protein-dependent pathway; instead, NCL knock-down reduces self-renewal ability and cell proliferation, upregulating p53 protein level and inducing cell apoptosis [78].

## 3. Human ESCs: Signaling Pathways Involved in Self-Renewal and Pluripotency

The TGF-β/Activin A/Nodal pathway has been shown to maintain the pluripotency state of hESCs by activating Smad2/3 [79,80,81]. Smad2/3 supports the self-renewal properties of hESCs through activation of Nanog [82,83]. Smad2/3 activates Nanog expression by binding the promoter of Nanog and inhibiting autocrine BMP signaling [82,84,85,86]. Inhibition of Activin A/Nodal induces differentiation by reducing gene expression of Nanog, Oct4, and Sox2. In the pluripotency state, Smad2/3 and Nanog suppress the expression of Smad-interacting protein 1 (SIP1) to inhibit neuroectodermal differentiation [87]. Furthermore, Smad2/3 can interact with the Jun N-terminal kinase (JNK)-JUN family and SnoN (SKIL) to block differentiation [88,89]. Importantly, low concentrations of Activin A (5 ng/mL) can sustain pluripotency and self-renewal of hESCs [90,91], whilst high concentrations of Activin A (50–100 ng/mL) decrease pluripotency and induce differentiation into endoderm [92]. In agreement with these finding, the chemokine (C-X-C motif) ligand 14 (CXCL14) has been shown to support the expression of self-renewal markers including Oct4 and Nanog by binding to the IGF-1R of hESCs [93]. Conditions like culturing hPSCs on simulated microgravity can enhance self-renewal properties by triggering the overexpression of pluripotent transcription factors, eventually leading to an upregulation of cell cycle regulators and reducing differentiation [94].

Generally, hESCs are cultured on Mouse Embryonic Fibroblast (MEF) or under feeder-free conditions such as on Matrigel in a medium containing basic fibroblast growth factor (bFGF) and Activin A or TGF-β. Basic Fibroblast Growth Factor (bFGF) signaling supports pluripotency, stemness, and hESCs self-renewal by activating the phosphatidylinositol 3-kinase (PI3K)/AKT pathway, inhibiting ERK activity, WNT signaling, and dephosphorylation of glycogen synthase kinase-3 (GSK3β) [95,96,97,98,99]. The PI3K/AKT signaling pathway regulates proliferation and pluripotency of ESCs through promoting gene expression of Nanog [100,101,102]. This pathway also upregulates the gene expression of c-Myc and Tbx3 via GSK3 inhibition [103]; instead, AKT inhibition by GSK690693, AKT inhibitor VIII, and AKT inhibitor IV, induces apoptosis and reduces self-renewal of hESCs and hiPSCs [104] (Figure 3).

Cyclin Dependent Kinase 1 (CDK1) is one of the main regulators of mitosis [105,106,107,108] and its downregulation has been shown to impair pluripotency and promote differentiation by suppressing the PDK1-PI3K/AKT pathway [109]. CDK1 is required for self-renewal of both mESCs [110] and hESCs [111], most likely because of its interaction with Oct4 [112,113,114]. Oct4 is a strategic transcription factor for the maintenance of pluripotency. During development, Oct4 is expressed in blastomers, the ICM, and epiblasts, and is then downregulated in the gastrulation stage [115]. Oct4 and Sox2 sustain hESCs self-renewal by upregulation of the Wnt-β Catenin antagonist SFRP2 and repression of the WNT trafficking protein WLS [116,117]. In agreement with these findings, Cooney and collaborators have demonstrated that germ cell nuclear factor (GCNF), an orphan nuclear receptor, suppresses the expression of Oct4 and modulates gene expression in hESCs [118].

Briefly, pluripotency and self-renewal of mESCs depend on BMP/Smad/Id and LIF/STAT3 signaling in culture, which is a hallmark of naïve pluripotency. In fact, mESCs have been shown to be able to maintain pluripotency and self-renewal without LIF, BMP4, or any external signaling by adding inhibitors of MAPK (ERK1/2 pathway) and glycogen synthase kinase 3 (so-called “2i medium”) [119]. Instead of LIF and 2i, hESCs require FGF/ERK and Activin/Nodal signaling to sustain self-renewal and pluripotency.

## 4. Epigenetic and Chromatin Modifications of Self-Renewal and Pluripotency in ESCs

Regulation of chromatin packaging through histone modification and DNA methylation plays an elemental role in maintaining self-renewal and pluripotency of ESCs. DNA methylation is a form of modification in which a methyl group is transferred onto cytosines in the CpG dinucleotide [120,121,122,123,124,125,126]. In this regard, there are two main histone modifications to regulate gene expression: histone H3 lysine 4 and 27 trimethylations (H3K4me3 and H3K27me3, respectively) [127,128]. H3K4me3 is related to active transcription [129], while H3K27me3 modification represents a negative regulator of gene expression and attracts chromatin inhibitors, thereby diminishing transcription [130]. During differentiation, H3K27me3 inhibits self-renewal and pluripotency. Genomic regions that present both histone modifications (H3K4me3 and H3K27me3) are known as bivalent domains [131,132,133,134]. These regions of the genome are essential for ESCs self-renewal and pluripotency. Recently, Nanog has been shown to repress differentiation genes by sustaining H3K27me3 in mESCs [135].

A member of the Ten-eleven translocation (Tet) family, Tet1, promotes gene expression of Oct4 by demethylation of CpG islands of promoter regions [136,137,138]. N^6^-methyladenosine (m^6^A) is a modification that occurs on mRNA and long noncoding RNA [139,140]. Wen at al reported that the Zinc Finger protein CCCH-Type Containing 13 (ZC3H13) is important for the localization of WTAP (WT1 Associated Protein), Virilizer (VIRMA: Vir Like m6A Methyltransferase Associated), and Hakai (CBLL1: Cbl Proto-Oncogene Like 1) in the nucleus to promote m6A methylation and to eventually modulate mESCs self-renewal [141]. DNA methyltransferase 3A and 3B (Dnmt3A, Dnmt3B)-deficient ES cells are not able to differentiate, but maintain self-renewal ability [142]. Other non-coding RNAs have also been implied in the regulation of the mechanisms underlying pluripotency and self-renewal [143,144,145,146,147,148,149,150,151,152].

Intriguingly, AICAR has been shown to sustain self-renewal of mESCs by altering the expression of epigenetic markers including Dnmt3a, Dnmt3b, Smarca2, Mbd3, and Arid1a [153]. Finally, Jia and colleagues reported that the undifferentiated embryonic cell transcription factor 1 (Utf1), the gene target of Oct4 and Sox2, promotes self-renewal and pluripotency of hESCs by inhibiting H3K27me3 modification and the polycomb repressive complex 2 (PRC2) [154].

## 5. Naïve Pluripotency

Up to now, two distinct states of pluripotency have been observed in the murine system: namely, naïve state and primed state. Specifically, “naïve” cells are present in an earlier developmental stage (the peri-implantation blastocyst), whereas “primed” cells exist in the later, post-implantation embryo. In mice, mESCs were shown to settle in an ICM-like state, termed “naïve pluripotency”, at E4.5. Mouse epiblast stem cells (mEpiSCs) are derived from post-implantation epiblast cells and named “primed” (from E5.5 onwards). hESCs present a primed state of pluripotency. For their self-renewal, mEpiSCs, like hESCs, depend on bFGF and Activin A/Nodal pathways that maintain the expression of OCT4. The stability of these states is controlled by endogenous and exogenous factors. For instance, culturing mESCs in medium containing primed factors including FGF-2 and TGFβ1/Activin A can convert the naïve state into the primed state and support the self-renewal of EpiSCs [155,156]. Interestingly, mESCs were converted into ground state of primed epiblast stem cells (rsEpiSC: region-selective EpiSC) by changing conditions from 2i/LIF to a medium containing FGF2 and a Wnt signaling inhibitor (IWR1); the converted cells exhibited properties of the primed state, including X chromosome inactivation [157].

Naïve and primed pluripotency show significant differences in growth, gene expression, cellular fitness, metabolism, and differentiation potential [158,159,160,161,162,163,164]. Both mESCs and hESCs express OCT3/4, SOX2, SSEA-3, GATA4; mESCs express different genes including SSEA-1, FoxD3, and LIF receptor complex (LIFR/IL6ST); they exhibit a dome-shaped colony morphology, karyotype stability, and two active X chromosomes in female cells. hESCs share a number of properties with murine naïve pluripotency while these features are absent in EpiSCs [165,166], suggesting that hESCs are less primed compared to murine EpiSCs, but they still have many features of EpiSCs; hESCs show a flat colony morphology and X chromosome inactivation in female cells, accumulation of H3K27me3, low ability to generate primordial germ cell-like cells (PGCLCs), and are identified by the expression of SSEA-4 (Stage-specific embryonic antigen 4), Podocalyxins TRA-1-81 and TRA-1-60, and Vimentin; they are also able to differentiate into trophoblast-like cells [167]. FGF/TGFβ signaling supports the inactivation of X chromosome and increases DNA methylation in the primed state of hPSCs [168,169].

During the last decade, much effort has been made to generate and evaluate human naïve pluripotency in vitro. Multiple protocols have been reported to generate human naïve pluripotency via conversion from primed PSC, reprogramming from somatic cells, and derivation from embryos [170,171,172,173,174,175,176,177,178]. Jacob Hanna and coworkers generated human naïve PSCs by ectopic expression of OCT4, Klf4, and Klf2 in a medium containing LIF and inhibitors of GSK3β and the ERK1/2 pathway [179]. Strikingly, combining 2i/LIF (2iL) and tamoxifen with forced expression of hormone-dependent STAT3-ER (ER, ligand-binding domain of the human estrogen receptor) reprogrammed hESCs from a primed state to naïve pluripotency through activation of STAT3 target genes [170]. Ohad Gafni and co-workers established a naïve human stem cell medium (NHSM) containing 2iLIF (2iL), P38i/JNKi, protein kinase C inhibitor (PKCi), Rho-kinase inhibitors (ROCKi), Activin, and FGF2, to capture and support human naïve PSCs [174]. In a similar study, Thorold Theunissen and collaborators identified five inhibitors targeting MEK, GSK3, BRAF, ROCK, and SRC together with LIF and two growth factors, i.e., FGF and Activin A (5i/L/FA), to support human naïve pluripotency and OCT4-ΔPE-GFP^+^ and transgene-free naïve hESCs [180]. The *Oct4*-ΔPE-GFP marker has been harnessed to discriminate naïve human PSCs from primed human PSCs [174,181]. Yet, *Oct4*-ΔPE-GFP reporter activity is not completely negative in primed hESCs and still includes a weak GFP activity; additionally, *Oct4*-ΔPE-GFP^+^ cells may still include ESCs utilizing *Oct4*-PE, since a mono-transgenic system is unable to discern between cells using only *Oct4*-DE and cells using both *Oct4*-DE and *Oct4*-PE, which may constitute an impure population of naïve PSCs. To overcome these issues, a dual reporter system for naïve and primed mouse PSCs has been generated using two fluorescent reporters, GFP and tdTomato (RFP), controlled by the *cis*-regulatory elements DE and PE, respectively: the expression of *Oct4*-ΔPE-GFP and *Oct4*-ΔDE-RFP was shown to accurately represent the expression of naïve and primed cells during the development of double transgenic mice [182]. Henceforth, this double transgenic system recapitulates the in vivo *Oct4* regulatory system, providing an exquisite tool for assessing the regulation of naïve and primed pluripotency while enabling the separation of pure populations of naïve and primed PSCs.

Overexpressing YAP, the Hippo pathway effector, in hESCs and iPSCs was found to enhance naïve pluripotency [177]. In these conditions, hPSCs show some naïve characteristics including reduced DNA methylation, upregulation of naïve pluripotency markers, nuclear localization of TFE3, in vitro generation of PGCs, and inhibition of MEK/ERK signaling. Nonetheless, there is still a gap between mouse and human naïve pluripotency. Recently, signaling pathway modulations have been reported to enhance human naïve pluripotency induction [183]. 

In order to enhance efficiency of human naïve pluripotency induction, Jacob Hanna’s research team engineered WIBR3 hESC lines with inducible ablation of METTL3 expression (Tet-OFF-METTL3 hESCs): however, these cells were maintained poorly in NHSM medium [183]. Therefore, these researchers modified some previous protocols showing that adding IWR1, WNT inhibitor (WNTi), increased the expansion of their Tet-OFF-METTL3 hESCs. They also observed that culturing the DPE-OCT4-GFP naïve pluripotency reporter line described by Theunissen et al. [181] in NHSM supplemented with TNKi, JNK/P38 inhibitor, and in absence of GSK3i, significantly increased human naïve pluripotency. Screening small molecules in NHSM medium revealed a defined FGF/TGF/Activin/MEF-independent and serum-free condition named human enhanced naïve stem cell medium (HENSM), that strongly supported naïve hPSCs in the absence of DNMT1, exogenous L-glutamine, and increased oxidative phosphorylation (OXPHOS) activity, presented a dome-like morphology, formed mature teratoma, showed a normal karyotype after extended passaging, differentiation into bona fide extra-embryonic trophoblast stem cells and extra-embryonic naïve primitive endoderm cells (PrE and nEND cells), upregulation of naïve pluripotency markers including Nanog, TFCP2L1, Klf4, Klf17, DPPA3 (also known as STELLA, STELLAR, and PGC7), DPPA5, DNMT3L, KHDC1L, and ARGFX, downregulation of DNA methylation from 81% to 66%, reactivation of both X chromosomes, and lack of immediate global loss of all imprinted genes compared to NHSM, 5i/L/FA, and t2iL-Go conditions [178,184]. In this protocol, WNT/β-Catenin inhibition supported human naïve pluripotency, unlike murine naïve pluripotency, as indicated using TNK inhibitors and β-Catenin KO cells in naïve pluripotency reporter lines; similarly the inhibition of Notch/RBPj was shown to induce the expression of human naïve pluripotency markers without inhibiting ERK. Khan et al. described a protocol that included FGFR, RAF, or ERK alongside the dual inhibition of MEK and ERK with PKC, ROCK, and TNK inhibitors and Activin A to support human naïve pluripotency [185]. 

Besides, Klf17 was shown to enhance naïve pluripotency by suppressing the transcription of MAPK3 and ZIC2 in hESCs [186]. Remarkably, recent studies have revealed a high gene-targeting efficiency of human naïve pluripotency by CRISPR/Cas9 system compared to primed pluripotency [187]. Therefore, human naïve pluripotency provides a better model to study embryogenesis and mechanisms of pluripotency [188,189,190]. Bi and colleagues reported alkaline phosphatase placental-like 2 (ALPPL2) as an important naïve-specific surface marker in human naïve PSCs: ALPPL2 is necessary for the maintenance and establishment of naïve pluripotency [191]. ALPPL2 preserved mRNA levels of the naïve pluripotency transcription factors including TFCP2L1 (Transcription Factor CP2 Like 1, also known as CRTR1 or LBP-9) and STAT3 (Signal Transducer and Activator of Transcription 3, also known as ADMIO1 and PRF) via interacting with the RNA-binding protein IGF2BP1 (Insulin Like Growth Factor 2 MRNA Binding Protein 1) [191]. Endogenous retroviruses (ERVs) are viral elements with long terminal repeats (LTRs) that can be derived from retroviruses. ERVs are active during early embryogenesis and are silenced transcriptionally in somatic cells. Klf5 interacts with and rewires Nanog to bind the naïve-specific LTR7Y sites, and facilitate expression of naïve and trophectoderm genes in naïve hESCs; in absence of Klf5, Nanog is bound to the primed-enriched LTR7 sites in primed hESCs [192,193,194].

Sox2 is a core pluripotency factor and its ablation in mESCs results in loss of self-renewal, pluripotency, and modification of ESCs into trophoblast-like stem cells [195,196]. In fact, Sox2 modulation increased naïve pluripotency plasticity. Tremble et al. reprogrammed Sox2-low iPSCs from Sox2-/- NSCs using retroviral factors: cMyc, Klf4, Oct4, and Sox2 (rMKOS) in defined culture conditions containing inhibitors of MEK/ERK and Gsk3β signaling supplemented with LIF (2il). Sox2-low iPSCs expressed naïve pluripotency markers (Nanog, Esrrb, Klf4, Oct4) and suggested that the low expression of Sox2 is sufficient to maintain a naïve molecular signature. Likewise, Sox2-low iPSCs differentiated into both embryonic (ectoderm, mesoderm, and endoderm) and extraembryonic lineage (trophoblast), in vitro as well as in vivo [39].

## 6. Controversial Issues

In recent years, significant advances have been made to convert the primed state of hPSCs into naïve pluripotency. Human naïve pluripotency has been shown to differentiate better than the primed state, which is associated with the accumulation of DNA methylation and epigenetic repressive marks in primed state. The Wang laboratory highlighted the role of Klf17 in promoting naïve pluripotency in hESCs [186], while Bayerl et al. reported that Klf17 is non-necessary for mouse naïve pluripotency [183]. Further investigations into species-specific functions of Klf17 are warranted.

Chromosome X reactivation in human naïve pluripotency observed in HENSM, NHSM, 5i/L/FA, and t2iL-Go, cannot yield inactivation upon hPSC differentiation [197]. So, it will be imperative to determine which conditions strongly protect long-term genomic integrity of naïve hESCs. Another crucial finding in naïve pluripotency is the higher expression of IGF1R and phosphorylation of AKT, which should be examined in depth. Inhibition of MEK signaling blocks self-renewal in the presence of IGF1 and Activin, but whether ERK is functionally involved remains unclear. It would be also noteworthy to assess the actual role of integrins in maintenance of pluripotency.

Additionally, there are some open challenges in stem cell therapy and stem cell directed differentiation, including the attempts to improve their properties in a manner similar to the in vivo setting. Future dedicated studies on pluripotent epiblast should clarify the functional role of precise signaling pathways during early development and delineate which ones are the most effective for in vitro-derived hESC lines. Culturing hPSCs and murine ESCs in mature conditions and the combination of in vitro and in vivo assays would be useful to resolve some remaining issues regarding the functional contribution of defined signaling pathways in pluripotency, and self-renewal of hESCs and mESCs, with the ultimate goal being to promote regenerative medicine.

## 7. Conclusions

ESCs are derived from ICM within blastocysts. They proliferate indefinitely and are able to differentiate into three embryonic germ layers, namely endoderm, mesoderm, and ectoderm. Both self-renewal and pluripotency are dependent on a finely tuned correlation and balance of signaling pathways, transcription factor programs, and epigenetic modifications. Understanding signaling pathways to control ESCs self-renewal and pluripotency could be extremely helpful to better delineate their clinical applications [198,199,200].

The transition between different states of pluripotency represents an exquisite model to painstakingly study the development of early human pre-implantation. Culturing human naïve PSCs is possible, and screening growth factors and small molecules might be useful for optimizing conditions to close the gap between mouse and human naïve pluripotency. In this review, we summarized the main molecular mechanisms and signaling pathways involved in self-renewal and pluripotency in mESCs and hESCs, and in capturing human naïve pluripotency from primed state of pluripotency. Of course, there are still many unanswered questions regarding the exact correlation between signaling pathways, transcription factor networks, and epigenetic modifications, and further studies in this sense are warranted.

## Figures and Tables

**Figure 1 ijms-24-08386-f001:**
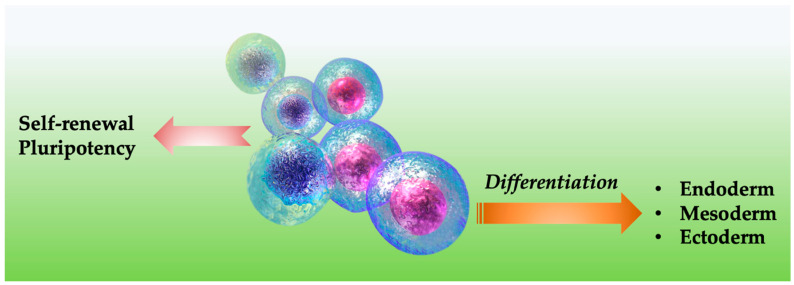
ESCs are able to differentiate into three embryonic germ layers: endoderm, mesoderm, and ectoderm.

**Figure 2 ijms-24-08386-f002:**
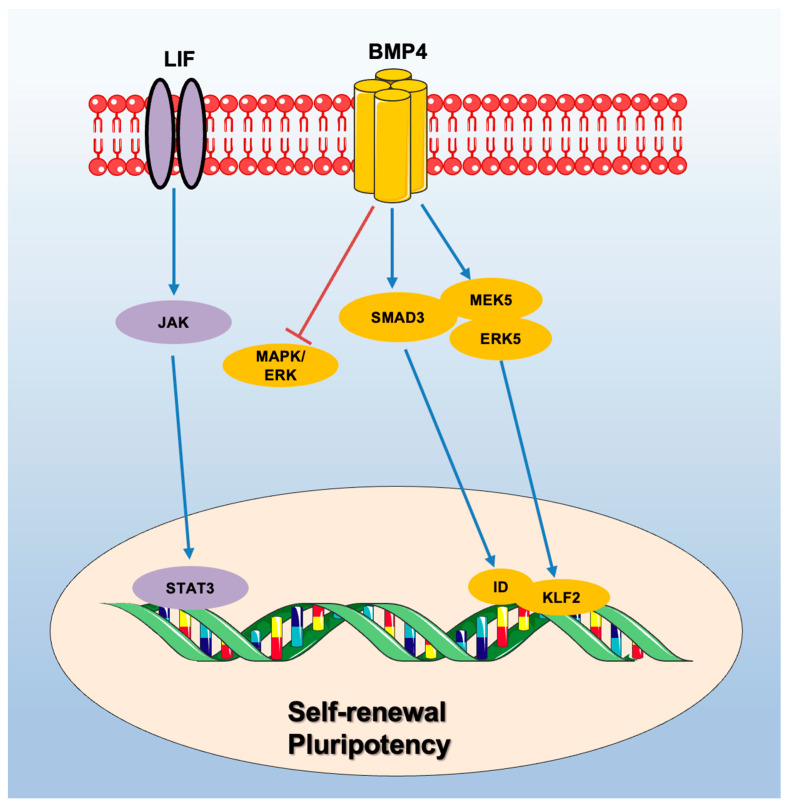
Major pathways involved in self-renewal and pluripotency in mouse ESCs.

**Figure 3 ijms-24-08386-f003:**
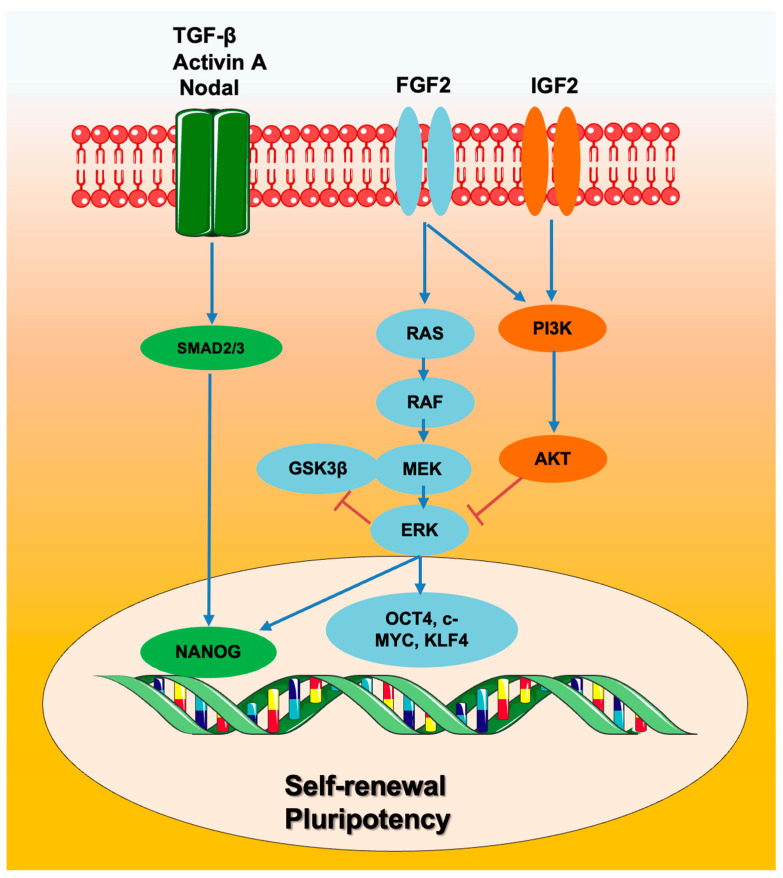
Main signaling pathways regulating self-renewal and pluripotency in human ESCs.

## Data Availability

Not applicable.

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
