# Peer review of "Molecular Mechanisms Underlying Pluripotency and Self-Renewal of Embryonic Stem Cells"

_ijms, 2023, doi:10.3390/ijms24098386_

Round 1

Reviewer 1 Report (New Reviewer)

The review submitted by Varzideh et al is very exhaustive in terms of the references cited and describes very meticulously the pathways involved in plutripotency and self-renewal. The part on primed and naive ESCs is particularly well explained and very interesting in mice and humans.

Author Response

The review submitted by Varzideh et al is very exhaustive in terms of the references cited and describes very meticulously the pathways involved in pluripotency and self-renewal. The part on primed and naive ESCs is particularly well explained and very interesting in mice and humans.

R: Thank you for your words of appreciation.

Reviewer 2 Report (New Reviewer)

The manuscript “Molecular mechanisms underlying pluripotency and self-renewal of embryonic stem cells” is focused on summarizing the scientific literature about the different factors and signaling pathways involved in the control of self-renewal and pluripotency in both mouse and human Embryonic Stem Cells (ESCs). This is surely a very important topic given the recent expansion of the scientific research in the field of human ESCs aimed at improving the use of human stem cells for therapeutic purposes. 

Although the topic is interesting, I think there are a number of concerns that should be addressed by the authors in order to make the manuscript suitable for publication:

-       Lane 219: beta III-tubulin is a neuronal marker. Sometimes a very little expression of specific differentiation markers can be found in pluripotent ESCs but is usually limited to RNA transcription. In any case, beta III-tubulin cannot be listed together with the other specific pluripotency markers such as SSEA-1 or FoxD3, this is totally incorrect and misleading.

-       The authors use the abbreviations mESCs and hESCs to distinguish mouse from human pluripotent stem cells. Many times, they use simply the abbreviation ESCs but I think that more attention should be paid, throughout the whole manuscript, to the use of the general abbreviation specifying or making explicit when a given scientific result/observation can be referred to both mouse and human stem cells. For example: 

a)     Lane 298-307: the authors described the role of Sox2 in regulating pluripotency but it is not clear at all whether they are talking about its involvement in regulating pluripotency in mouse or in human stem cells, or both.

-       Lane 245: “OCT4-ΔPE-GFP+ and transgene-free naïve hESCs”, please specify in more detail, what is the OCT4-ΔPE-GFP+ cell line?

-       Lane 254: again, what is the “Tet-OFF-METTL3 hESCs”?

-       Lane 292-297: the paragraph is all about conversion of naïve mESCs to a primed state but appears to be quite unlinked to the previous and the following discussion. Maybe, it should be moved to the first paragraph, at the beginning of the section 5. Naïve Pluripotency, where the primed state in human and mouse ESCs is discussed in detail. 

-       Lane 186-187: the expression “Expression of ectodermal and endodermal genes in ASH2L-/- knockout mESCs was low compared to wild-type mESC” is totally incorrect: the reduction in endodermal and ectodermal gene expression has been observed in mouse (differentiating) Embryo Bodies (EBS) not in mESCS that do not express, of course, endodermal and ectodermal genes. Beside this, the role of ASH2L in mESCs is more complex than described since there is a compensation effect by two ASH2H isoforms.

 As a general observation, I suggest a more careful reading/editing of the manuscript from the point of view of the language: there are several spelling and grammar mistakes, in addition to some truncated sentences (missing words/verbs) and some too colloquial expressions. For example: 

a)     Lane 156: “CDK-1 is highly expressed and regulates human embryonic…” what? Missing something.

b)    Lane168: “incoporation” should be “incorporation”. In this specific case, however, I think the use of this word is incorrect, it is not an “incorporation” but rather an “addition” to the colture medium.

c)     Lane 214, 217 and others: the use of “prime” is incorrect, “primed” should be used.

d)    Lane 234: “derivation” should replace “deviation”.

e)    Lane 220: “pexhibit” should be “exhibit”.

Author Response

-We agree with this Reviewer and we have removed beta III-tubulin.

-We have revised the use of mESC and hESC; specifically, in Lane 298-307 we clarify that we were referring to mESC.

-We apologize for the lack of clarity: we now better describe OCT4-ΔPE-GFP+ and Tet-OFF-METTL3 hESCs.

-The paragraph in Lane 292-297 has been moved to the first paragraph at the beginning of the section 5, as requested. 

-We agree and we removed lanes 186-187.

Errors listed by this Reviewer in points a-e have been rectified.

This manuscript is a resubmission of an earlier submission. The following is a list of the peer review reports and author responses from that submission.

Round 1

Reviewer 1 Report

The review entitled “Molecular mechanisms underlying pluripotency and self-renewal of embryonic stem cells” by Varzideh et al. gives a short summary of the current state of the topic. The manuscript is well written. 

The manuscript is separating human and mouse embryonic stem cell work and it would be of interest to add a paragraph addressing the differences and explanations therefore in human and mouse embryonic stem cell signalling.  

typo/grammar line 129. Isn't it fibroblasts?

Reviewer 2 Report

This review article discusses the molecular signals that regulate self-renewal and stemness in mouse and human embryonic stem cells (mESCs and hESCs). However, most of the signaling pathways described in the content of this review article are early established and well-known, including regulation of self-renewal and potency via TGF-β/Activin A/Nodal in hESCs, via LIF/BMP4 signaling in mESCs, or through specific epigenetic states such as DNA methylation. Overall, the content of this review article does not provide sufficient new findings in the field. There have been many good review articles in this field, but the depth and breadth of this article are far from them.

Reviewer 3 Report

The article is devoted to the molecular mechanisms of pluripotency and self-renewal in mouse and human embryonic stem cells. The review certainly shows the authors' good knowledge in this matter.

Nevertheless, the review is more like a chapter in a textbook than a critical literature review. The diagrams shown in the figures are too schematic. Practically nothing has been said about naive pluripotency, which is a large part of modern research in the field.

The most important thing is that there is no opinion of the authors in the review, a section where controversial or unresolved issues would be discussed. This leterature review will certainly be useful to students, but I am not sure that it will arouse the interest of specialists in the field of developmental biology or in the field of pluripotency studies. I don't see enough novelty and originality in this article.